# *MET* mutation causes muscular dysplasia and arthrogryposis

Hang Zhou[1,2,3,4,‡], Chengjie Lian[1,2,3,4,‡], Tingting Wang[1], Xiaoming Yang[1], Caixia Xu[5], Deying Su[1], Shuhui Zheng[5], Xiangyu Huang[6], Zhiheng Liao[1], Taifeng Zhou[1], Xianjian Qiu[7], Yuyu Chen[1], Bo Gao[7], Yongyong Li[5], Xudong Wang[7], Guoling You[8], Qihua Fu[8], Christina Gurnett[9,10,11], Dongsheng Huang[7] & Peiqiang Su[1,2,3,4,†,*] iD

## Abstract

Arthrogryposis is a group of phenotypically and genetically heterogeneous disorders characterized by congenital contractures of two or more parts of the body; the pathogenesis and the causative genes of arthrogryposis remain undetermined. We examined a four-generation arthrogryposis pedigree characterized by camptodactyly, limited forearm supination, and loss of myofibers in the forearms and hands. By using whole-exome sequencing, we confirmed *MET* p.Y1234C mutation to be responsible for arthrogryposis in this pedigree. *MET* p.Y1234C mutation caused the failure of activation of MET tyrosine kinase. A *Met* p.Y1232C mutant mouse model was established. The phenotypes of homozygous mice included embryonic lethality and complete loss of muscles that originated from migratory precursors. Heterozygous mice were born alive and showed reduction of the number of myofibers in both appendicular and axial muscles. Defective migration of muscle progenitor cells and impaired proliferation of secondary myoblasts were proven to be responsible for the skeletal muscle dysplasia of mutant mice. Overall, our study shows *MET* to be a causative gene of arthrogryposis and *MET* mutation could cause skeletal muscle dysplasia in human beings.

**Keywords** arthrogryposis; MET; muscle development; muscular dysplasia; whole-exome sequence
**Subject Categories** Genetics, Gene Therapy & Genetic Disease; Musculoskeletal System

## Introduction

Arthrogryposis is a group of disorders characterized by congenital joint contractures that mainly involve two or more areas of the body (Bamshad *et al*, 2009; Bayram *et al*, 2016) and affects approximately 1 in 3,000 newborns (Bayram *et al*, 2016). Structural and functional disorders of skeletal muscles are the most common reason for arthrogryposis (Toydemir *et al*, 2006). Arthrogryposis is a group of disorders with high clinical and genetic heterogeneity. Variants in more than 220 genes have been found to be associated with arthrogryposis (Narkis *et al*, 2007; Drury *et al*, 2014; Hunter *et al*, 2015; Bayram *et al*, 2016). However, the molecular etiology still remains unclear in a large number of cases of arthrogryposis. Further studies to identify causative genes and pathogenic mechanisms are needed.

MET belongs to the receptor tyrosine kinase family, and it is encoded by the MET proto-oncogene, receptor tyrosine kinase (*MET*, MIM:164860) gene. The biological effects exerted by MET are triggered by the stimulation of its only ligand, hepatocyte growth factor (HGF; Trusolino *et al*, 2010). Upon ligand binding, MET is autophosphorylated on tyrosine (Y)-1234/1235 in the activation loop of the MET catalytic domain. Y-1234/1235 phosphorylation is required for the activation of MET kinase and subsequent phosphorylation of other tyrosine sites of MET, including Y-1003 in the juxtamembrane domain, and Y-1349 and Y-1356 in the carboxyl terminus (Sangwan *et al*, 2008). Phosphorylated Y-1349 and Y-1356

1   Department of Orthopaedic Surgery, First Affiliated Hospital, Sun Yat-sen University, Guangzhou, Guangdong, China
2   Guangdong Provincial Key Laboratory of Orthopedics and Traumatology, First Affiliated Hospital, Sun Yat-sen University, Guangzhou, Guangdong, China
3   Guangdong Province Center for Peripheral Nerve Tissue Engineering and Technology Research, Guangzhou, Guangdong, China
4   Guangdong Province Engineering Laboratory for Soft Tissue Biofabrication, Guangzhou, Guangdong, China
5   Research Centre for Translational Medicine, First Affiliated Hospital, Sun Yat-sen University, Guangzhou, Guangdong, China
6   Department of Stomatology, Nanfang Hospital, Southern Medical University, Guangzhou, Guangdong, China
7   Department of Spine Surgery, Sun Yat-sen Memorial Hospital, Sun Yat-sen University, Guangzhou, Guangdong, China
8   Department of Laboratory Medicine, Shanghai Children's Medical Center, Shanghai Jiao Tong University School of Medicine, Shanghai, China
9   Department of Orthopaedic Surgery, Washington University, St. Louis, MO, USA
10  Department of Neurology, Washington University, St. Louis, MO, USA
11  Department of Pediatrics, Washington University, St. Louis, MO, USA
    *Corresponding author. Tel: +862087755766 6236; E-mail: supq@mail.sysu.edu.cn
    †Present address: Department of Orthopaedic Surgery, First Affiliated Hospital, Sun Yat-sen University, Guangzhou, Guangdong, China
    ‡These authors contributed equally to this work

serve as multifunctional binding sites for GAB1, GRB2, PI3K, and other downstream substrates (Birchmeier et al, 2003). The HGF-MET signal plays a vital role in regulating the development of skeletal muscle, placenta, and liver during embryogenesis (Birchmeier et al, 1997; Haines et al, 2004; Ueno et al, 2013). In skeletal muscle development, MET has been demonstrated not only to be crucial for the migration of muscle progenitor cells into the limbs, tongue, and diaphragm, but also to be necessary for the proliferation of secondary myoblasts in the trunk (Maina et al, 1996).

Herein, we recruited a rare four-generation Chinese arthrogryposis pedigree with only upper limb involvement, and we found the MET c.A3701G (p.Y1234C; Refseq NM_000245.2) mutation to be responsible for the pathogenesis of arthrogryposis in this pedigree. MET p.Y1234C mutation was shown to cause the dysfunction of phosphorylation and tyrosine kinase activity of MET in vitro. We established a Met c.A3695G (p.Y1232C; Refseq NM_008591.2) mutant mouse model, and the defective migration of myogenic progenitor cells and impaired proliferation of secondary myoblasts were demonstrated to be responsible for the disturbed muscle development.

# Results

## Clinical presentation of patients from a large arthrogryposis family

A four-generation Chinese family presented with completely penetrant, autosomal dominant arthrogryposis characterized mainly by camptodactyly (Fig 1A). All patients in this family had camptodactyly, and seven patients had camptodactyly, absent flexion crease, and limited forearm supination (Fig 1B; Table EV1). Signs of lower limb, and facial and spinal involvement were absent. Since interphalangeal joints and carpal joints were both affected in seven individuals, a diagnosis of arthrogryposis involving only the upper limbs was made.

Subject IV:7 is a patient with unilateral camptodactyly, absent flexion crease, and limited forearm supination. Severe pronator quadratus aplasia of affected forearm was observed through magnetic resonance imaging (MRI; Fig 1C). For the palmar muscles, loss of lumbricalis and interosseous muscles of fifth finger of affected side was found (Fig 1D and E). Subject IV:8 is a patient with severe bilateral camptodactyly, absent flexion crease, and limited forearm supination. MRI scan showed increased epimysial fat among muscle compartments (Fig 1F), complete loss of thenar eminences, the radial lumbricalis, and interosseous muscles of both hands (Fig 1G and H). The lumbricalis muscle of subject IV:8 showed varying fiber size and more centrally located nuclei than control lumbricalis muscle from an age- and gender-matched person without muscular dysplasia (Fig EV1A). No bone abnormality was observed in arthrogryposis patients of this pedigree (Fig EV1B). Overall, a diagnosis of arthrogryposis involving only the upper limbs was made, and muscular dysplasia was observed in the affected forearms and hands of these patients.

## Whole-exome sequencing identified MET as a disease-causing gene of arthrogryposis

To identify arthrogryposis-predisposing variants, whole-exome sequencing was initially performed on four affected individuals and one healthy member of this arthrogryposis pedigree (Appendix Table S1). As previously reported (Gao et al, 2017), we annotated and filtered variants, and kept variants that were novel in dbSNP. Polyphen-2, Mutation Taster, and Genomic Evolutionary Rate Profiling (GERP) were then used to predict the potential functional effects of these mutations, which yielded two candidate SNVs, c.A3701G in the MET; c.G2074A (Refseq NM_006019) in TCIRG1 (MIM:604592). By using Sanger sequencing, we excluded the SNV on TCIRG1 because MET c.A3701G turned out to be the only one which co-segregated with disease phenotypes in this family (Fig 1I, Appendix Table S2).

## MET p.Y1234C mutation caused dysfunction of the phosphorylation and tyrosine kinase activity of MET

The influence of p.Y1234C mutation on the function of MET was studied (Fig EV2A), and HGF treatment was shown to be unable to phosphorylate the Y-1234/1235, Y-1349, and Y-1356 sites of mutant MET receptor (Fig EV2B–D), suggesting MET mutation impaired the activation of MET receptor. Moreover, the tyrosine kinase activity of mutant MET was shown to decrease dramatically (Fig 1J).

## Met mutation resulted in the reduction of limb myofibers in transgenic mouse model

To determine the mechanism by which MET mutation causes arthrogryposis, a Met p.Y1232C (which was identical to p.Y1234C in human beings) mutant mouse model was constructed. No homozygous newborns were found. The ratio of homozygous embryos started to decline since E14.5, and E16.5 was the latest time that homozygous embryos could survive, which was consistent with Met null mutants (Schmidt et al, 1995). The failure of placental development in homozygotes might be responsible for the death of embryos in utero (Ueno et al, 2013).

Heterozygotes were smaller than wild-type individuals at birth (Fig 2A and B). Compared with wild-type newborns, the mean number of myofibers of paraspinal muscles, forelimbs, hindlimbs, and hands of heterozygotes reduced by 14, 55, 29, and 93%, respectively, while the foot muscles remained normal (Fig 2C–H). To figure out whether the myofibrils were affected, the gastrocnemius of wild-type and heterozygous newborns was tested with transmission electron microscope (TEM). However, no abnormality of the structure of myofibrils was found (Fig 2I).

## Met mutation affected the migration of muscle progenitor cells

To determine how Met mutation affected muscle development, its effect on muscle progenitor cells' migrating out of dermomyotome was examined firstly. Using in situ hybridization, expression of Pax3 (MIM: 606597) and Met was assessed in embryonic limbs and dermomyotome at the end of migration (E10.5). In dermomyotome, heterozygous and homozygous embryos showed more Pax3-positive (Pax3[+]) and Met-positive (Met[+]) cells than wild types (Figs 3A and EV3). Pax3[+] and Met[+] cells were absent from homozygous limbs, while the number of Pax3[+] and Met[+] cells was markedly lower in heterozygous limbs, suggesting Met mutation impaired muscle progenitor cells' migration out of dermomyotome to the limb (Figs 3A and EV3).

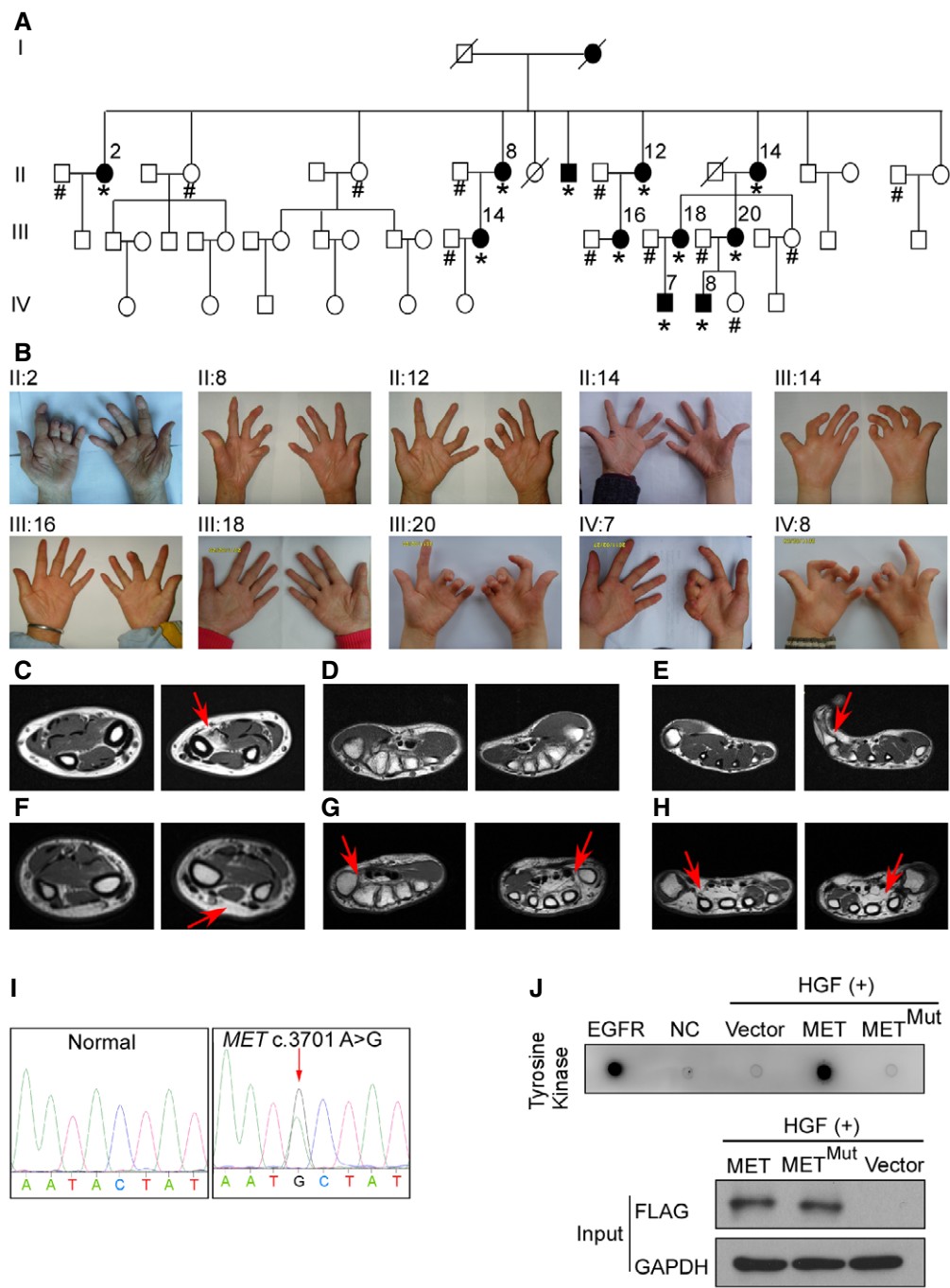

**Figure 1.   MET p.Y1234C mutation caused arthrogryposis in a four-generation Chinese family.**

A   The MET p.Y1234C mutation segregated with disease phenotypes in the arthrogryposis family. Filled symbols denote affected individuals, open symbols indicate unaffected individuals, and symbols with slashes represent decreased individuals. Asterisks indicate a mutation is present, # means wild-type.

B   Phenotypes of affected individuals. Camptodactyly, absent flexion crease, and limited forearm supination were observed.

C–E   T1-weighted MRI scan on upper limbs of subject IV:7. (C) The pronator quadratus absence of affected side was indicated by a red arrow. (D) No difference was found in palmar muscles. (E) Loss of lumbricalis and interosseous muscles of fifth finger of affected side was indicated by a red arrow.

F–H   T1-weighted MRI scan on upper limbs of subject IV:8. (F) Increased epimysial fat was indicated by a red arrow. (G) Completely loss of thenar eminences of both hands was indicated by red arrows. (H) Loss of radial lumbricalis and interosseous muscles of both hands was indicated by red arrows.

I   The MET variants by Sanger sequencing were indicated by a red arrow.

J   293T cells were transfected with FLAG-tagged MET/MET$^{Mut}$/Vector plasmids, and 48 h post-transfection, cells were treated with 10 ng/ml recombinant human HGF for 1 h. Then, MET/MET$^{Mut}$ protein purification and tyrosine kinase assay were conducted. Western blot pictures representative of $n = 3$ experiments. MET$^{Mut}$ means p.Y1234C mutant MET. EGFR means epidermal growth factor receptor and serves as a positive control. NC means negative control.

Source data are available online for this figure.

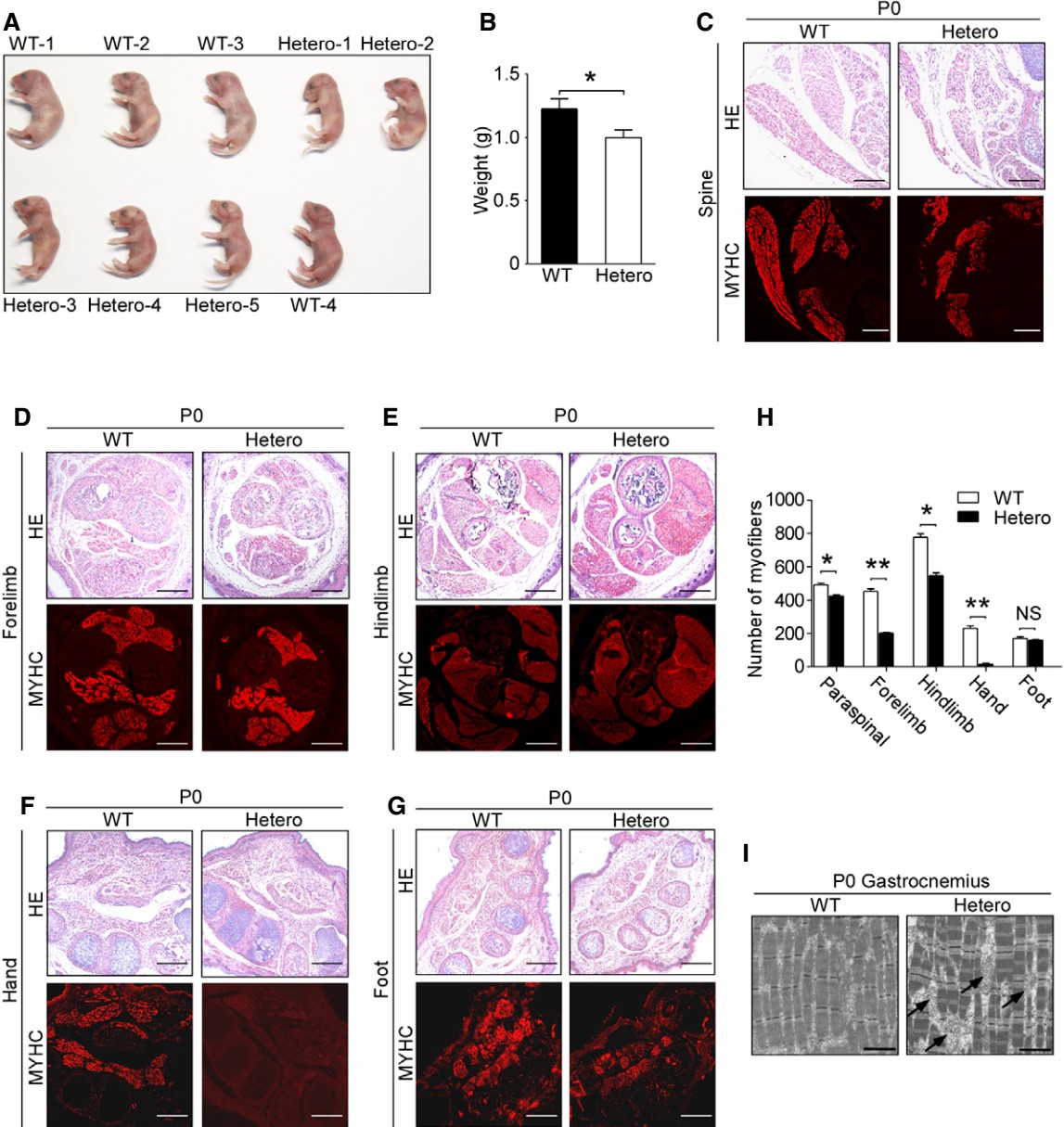

**Figure 2. *Met* mutation caused the reduction of both appendicular and paraspinal muscles of P0 heterozygotes.**

A    Gross appearance of wild-type and heterozygous newborns.

B    Graph showing weight of P0 mice. Bars show mean ± SD. Sample size: WT (*n* = 4) and Hetero (*n* = 5). *P < 0.05, by one-way ANOVA and followed by Dunnett's *post hoc* test.

C–G    Sections of spine, forelimbs, hindlimbs, hand, and foot from wild types and heterozygotes at P0 were conducted with HE staining and immunofluorescence staining using anti-myosin heavy chain antibody. Scale bars, 100 μm.

H    The mean number of myofibers in the muscles of the spine, forelimb, hindlimb, hand, and foot was qualified. Bars show mean ± SD. NS means no statistic significance. *P < 0.05, **P < 0.01, by two-tailed independent Student's *t*-test.

I    Transmission electron microscope analysis of gastrocnemius from wild-type and heterozygous newborns. Black arrows denote mitochondria. Scale bars, 2 μm. WT means wild types, Hetero means heterozygotes.

Data information: In (C–H), *n* = 3.

### *Met* mutation had no effect on primary myogenesis of embryonic muscle development

The decreased number of muscle fibers in axial muscles of P0 heterozygotes suggested that *Met* mutation might also affect the

myoblasts which do not undergo migration. The body size and weight of E14.5 (the very end of primary myogenesis) homozygous embryos were lower than that of wild types (Figs 3B and EV4A). For muscles that derive from migratory precursors, remarkable decrease of muscle fibers was observed in heterozygotes, while the

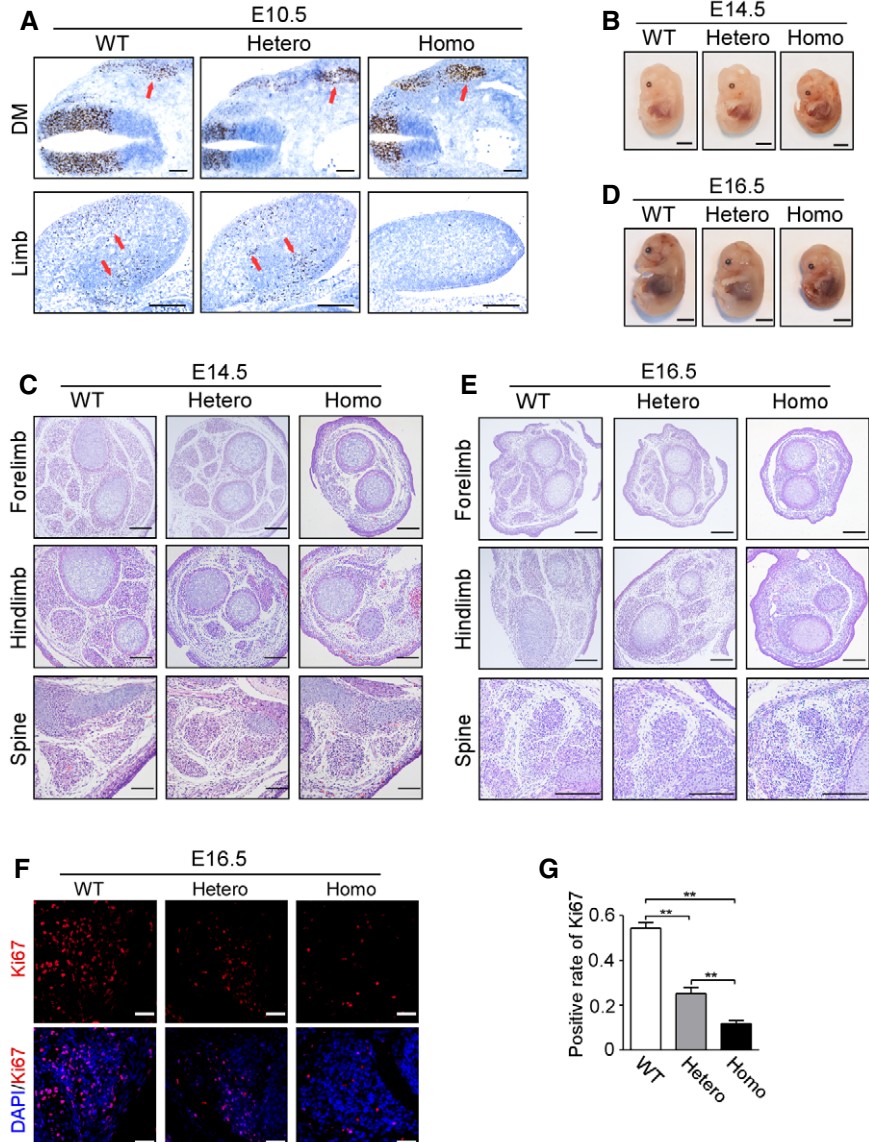

**Figure 3.  *Met* mutation led to the defects in the migration of muscle progenitor cells and impaired proliferation of secondary myoblasts.**

A   *In situ* hybridization of E10.5 embryos using *Pax3* probe. *Pax3* expression (brown signal) was observed in limb bud and dermomyotome (DM), respectively. Cells labeled with *Pax3* were indicated by a red arrow. Scale bars, 200 μm.
B   Gross appearance of E14.5 embryos of indicated genotype. Scale bars, 2.5 mm.
C   HE staining of forelimb, hindlimb, and paraspinal muscle of E14.5 embryos. Scale bars, 100 μm.
D   Gross appearance of E16.5 embryos of indicated genotype. Scale bars, 2.5 mm.
E   HE staining of forelimb, hindlimb, and paraspinal muscle of E16.5 embryos. *n* = 3, scale bars, 200 μm.
F   Anti-Ki67 antibody was used to label proliferative myoblasts (red fluorescence) with DAPI-labeled nuclei (blue fluorescence) in paraspinal muscle. Scale bar, 25 μm.
G   Bar graph showing statistical analysis of positive rate of Ki67-labeled nuclei, *n* = 3 with more than 150 cells analyzed per *n*, **P* < 0.01, by chi-square test (χ² test). Bars show mean ± SD. WT means wild types, Hetero means heterozygotes, and Homo means homozygotes.

homozygotes showed a complete loss of muscle fibers in these areas (limbs, front tongue, and diaphragm, Figs 3C and EV4B). For muscles that do not originate from the migratory precursors, no obvious difference was observed among three genotypes at E14.5 (paraspinal muscle and intercostal muscle, Figs 3C and EV4B). To evaluate the proliferation and apoptosis of primary myoblasts, Ki67 staining and TUNEL assay were conducted in paraspinal muscle, and no significant difference was found among all three genotypes

(Fig EV4C–F), suggesting that *Met* mutation had no effect on primary myogenesis.

## *Met* mutation suppressed proliferation of myoblasts during secondary myogenesis

To reveal the effect of *Met* mutation on secondary myogenesis, appendicular and axial muscles of E16.5 embryos (the latest time

that homozygotes could survive and also the late stage of secondary myogenesis) were studied. The body size and weight of E16.5 embryos decreased in heterozygotes and homozygotes (Figs 3D and EV5A). HE staining showed that compared to wild types, the mean myofiber numbers in heterozygotes decreased by 46% in forelimb and 45% in hindlimb, respectively, and a complete absence of myofiber was observed in homozygous limbs (Figs 3E and EV5B). HE staining revealed a graded reduction in paraspinal muscle fibers, progressively more severe in homozygotes than in heterozygotes (Figs 3E and EV5B).

To make it clear whether *Met* mutation affected apoptosis or proliferation of secondary myoblasts, TUNEL assay and Ki67 staining were performed in paraspinal muscle of E16.5 embryos. No significant difference in the TUNEL-positive rate was found among all three genotypes (Fig EV5C and D). However, in heterozygous and homozygous embryos, there was a reduction in Ki67-positive rate of 58 and 81%, respectively, relative to wild types (Fig 3F and G), indicating that a defect in proliferation of myoblasts was the reason for the impaired secondary myogenesis.

## Discussion

In the present study, *MET* p.Y1234C mutation was found to cause arthrogryposis in a four-generation pedigree. *In vitro* study showed that *MET* p.Y1234C mutation resulted in the failure of phosphorylation and loss of tyrosine kinase activity of MET receptor. A *Met* p.Y1232C transgenic mouse model was established, and defective migration of muscle progenitor cells and impaired proliferation of secondary myoblasts were detected, which was in accordance with previous study (Maina *et al*, 1996).

In our study, heterozygous *Met* p.Y1232C mutant mice also showed reduction of myofibers' number in both appendicular and axial muscles. Given that various heterozygous *Met* loss-of-function mutant mice did not show any abnormal phenotype (Maina *et al*, 1996, 2001; Sachs *et al*, 2000), the phenotype of *Met* p.Y1232C heterozygotes is likely to be caused by a dominant negative effect rather than by haploinsufficiency. Since Y-1232 is a crucial phosphorylation site in the MET kinase domain and MET activation depends on receptor dimerization in response to ligand binding (Trusolino *et al*, 2010), the p.Y1232C mutant might form non-functional dimers with the wild-type MET protein, resulting in impaired HGF-MET signaling. A similar effect has been previously described with a kinase-domain-truncated mutant MET (Furge *et al*, 2001; Long *et al*, 2003).

In P0 heterozygous mice, there was a complete absence of the intrinsic muscles of the hand, while the intrinsic muscles of the foot stayed normal, which was in accordance with the phenotypes of our arthrogryposis patients. Meanwhile, the reduction of myofibers in forelimb was more severe than that in hindlimb. One possible explanation for this inconsistency is that *MET* plays a predominant role in the early period of embryonic muscle development. As reported, *MET* was indispensable to the delamination of muscle progenitor cells from dermomyotome at the very beginning of migration. In *Met*$^{-/-}$ mice, the muscle progenitor cells failed to delaminate from the dermomyotome, and all the muscles that derived from migration failed to form as a result (Schmidt *et al*, 1995). Other genes, such as *Lbx1* and *Pax3*, are more important in regulating the migration of muscle precursor cells into the limbs during later periods of

migration (Relaix *et al*, 2004; Masselink *et al*, 2017). In homozygous *Lbx1* null mutant mice, limb muscle precursor cells could delaminate from dermomyotome normally but failed to migrate into the limb, which led to the loss of appendicular muscles (Gross *et al*, 2000). Since forelimb buds (E9.0–E9.5) formed earlier than hindlimb buds (E9.5–E10) during embryonic muscle development, it can be inferred that *Met* mutation caused severer phenotypes in forelimbs than hindlimbs because it mainly affects the early stage of migration. Although the reduction of myofibers in the extensor side of forelimbs was severer than that in the flexor side in some *Met* mutant mice models constructed before (Maina *et al*, 1996; Sachs *et al*, 2000), we did not find such differences in *Met* p.Y1232C mutant mice. It is also noteworthy that although *Met* p.Y1232C mutant mice recapitulated the phenotypes of muscular dysplasia in arthrogryposis patients, neither heterozygous nor homozygous mice showed contracture of distal joint. In the arthrogryposis family we examined, all patients with *MET* p.Y1234C mutation had camptodactyly, which was caused by the loss of the intrinsic muscles of the hands. Similar to these patients, *Met* c.A3695G mutant mice also showed loss of intrinsic muscles of the hands. The reason why mutant mice had no camptodactyly may be that the digits of mice are relatively short, so contracture is not readily visible. However, instead of contracture at the digits, the whole paw flexion at the wrist was observed in *Met* p.Y1356F mutant homozygotes (Maina *et al*, 1996).

It has been reported that SNVs in the *MET* gene, causing lowered MET expression, increase susceptibility to autistic spectrum disorders (ASD) in European and North American populations (Campbell *et al*, 2006). Mental evaluation by a psychiatrist of our arthrogryposis patients revealed no sign of ASD, possibly owing to the difference in genetic background among populations.

To the best of our knowledge, this is the first study to report *MET* as a causative gene of arthrogryposis. Though several mouse models have been established to study the role of *MET* on the development of skeletal muscle, the present study is the first to demonstrate a direct relationship between *MET* mutation and skeletal muscle dysplasia in arthrogryposis patients.

## Materials and Methods

### Subjects

Patients were recruited and evaluated in the First Affiliated Hospital of Sun Yat-sen University. Patients provided a detailed medical history, received physical examinations and mental evaluation, and underwent standard posterior–anterior plain X-rays of both the hands and the feet. MRI, electromyogram, and blood biochemical examination were conducted in subject IV:7. Histological analysis of lumbricalis and MRI were conducted in subject IV:8, and the lumbricales of an age- and gender-matched person with a severe hand injury served as normal control.

### Genetic studies

Exome sequences were enriched with an Agilent SureSelect Human All Exon V5 Kit (Agilent Technologies). Sequences were generated on a HiSeq PE150 (Illumina). Base calling was performed, and raw sequencing read files were generated in FASTQ format.

Subsequently, the sequenced reads were aligned to the reference human genome (NCBI Build 37, hg19). SeattleSeq Annotation 150 (version 9.10) was used to perform the annotation. Various data-bases, including dbSNP database, 1000 genomes, Mutation Taster, Polyphen-2, and SIFT, were used to predict and filter mutations. Data analysis was performed as given below: Exome capture was performed using an Agilent SureSelect Human All Exon Kit and sequences were generated on HiSeq PE150. After mapping to the human reference genome (NCBI Build 37, hg19), an average of 99.3% of reads were mapped and a sequencing depth per target base of 168 times on average was provided, with at least 20 times for 99% of bases. More than 154,876 SNVs per individual were identified. We first chose variants that were shared by the four affected individuals but not present in the unaffected individual. We found 5,350 such variants. Next, to identify potentially pathogenic variants, we anno-tated those variants and filtered out synonymous variants, non-coding variants, intergenic variants, and variants located in introns, retaining those affecting splice sites. We then parsed a total of 1,441 variants and kept only those that were novel in dbSNP.

### Immunoprecipitation

Flag-tagged MET, p.Y1234C mutant MET (MET$^{Mut}$), and Vector plas-mids were expressed in a 293T cell expression system. Forty-eight hours post-transfection, cells were treated with 10 ng/ml recombi-nant human HGF (PeproTch, Catalog No. 100-39H-25) for 1 h. As previously reported (Lian et al, 2016; Liu et al, 2018), lysates were prepared from $5 \times 10^7$ 293T cells transfected with indicated plas-mids using RIPA Lysis Buffer (Beyotime, Catalog No. P0013D). Lysates were incubated with 20 μl anti-Flag affinity agarose (Sigma-Aldrich, Catalog No. A4596) overnight at 4°C. Beads containing affinity-bound proteins were washed seven times with 5 ml wash buffer (300 mM NaCl, 20 mM HEPES, 1 mM EDTA, 1 mM EGTA, 2% glycerol, pH 7.4, and 0.1% NP-40) and collected by centrifuga-tion. Samples were subjected to SDS–PAGE and immunoblotting analysis after the addition of 30 μl of sample buffer [62 mM Tris–HCl, 1.25% (w/v) SDS, 10% (v/v) glycerol, 3.75% (v/v) mercaptoethanol, and 0.05% (w/v) bromophenol blue, pH 6.7] and denaturation. The following antibodies were used: 1:1,000 dilution of anti-Phospho-Met (Tyr1234/1235) antibody (CST, Catalog No. #3077), 1:1,000 dilution of anti-Phospho-Met (Tyr1349) antibody (Abcam, Catalog No. ab68141), 1:500 dilution of anti-Phospho-Met (Tyr1356) antibody (Abcam, Catalog No. ab73992), 1:2,000 dilution of anti-GAPDH antibody (Proteintech, Catalog No. 60004-1-lg), and 1:1,000 dilution of anti-FLAG antibody (Sigma-Aldrich, Catalog No. F1804).

### Tyrosine kinase assay

Flag-tagged MET/MET$^{Mut}$/Vector plasmids were expressed in a 293T cell expression system. Forty-eight hours post-transfection, cells were treated with 10 ng/ml recombinant human HGF for 1 h. MET/MET$^{Mut}$ protein purification and tyrosine kinase assay were conducted accord-ing to the operations manual (Sigma-Aldrich, Catalog No. CS0730).

### Generation of CRISP/Cas9 Met mutant mice

The mouse Met gene (GenBank accession number: NM_008591.2; Ensembl: ENSMUSG00000009376) is located on mouse chromosome

6, and human MET c. A3701G is identical to Met c.A3695G in mouse gene. Twenty-two exons have been identified, with the ATG start codon in exon 3 and TGA stop codon in exon 22. The Tyr1232 is located on exon 20. Exon 20 was selected as a target site. MET gRNA targeting sequencing 5′-GCTTGGCACCCGTCTTGTTGTGG-3′ and donor oligo were designed. The Tyr1232Cys (TAC to TGC) mutation sites in donor oligo were introduced into exon 20 using homology-directed repair. A silent mutation (GTC to GTA or ACG to ACT) was also introduced to prevent the binding and recutting of the sequence by gRNA after homology-directed repair. Cas9 mRNA, gRNA generated by in vitro transcription, and donor oligo were co-injected into fertilized eggs for KI mouse production.

### Histological study

Tissues from E14.5, E16.5, and P0 mice were dissected and fixed in 4% paraformaldehyde overnight, dehydrated, and embedded in paraffin. Sections for histological analysis were rehydrated and stained with hematoxylin-eosin. Immunofluorescence was performed with Histostain-Plus Kit (ZSGB-BIO, Catalog No. SPN-9002). Primary antibodies included: 1:400 dilution of anti-myosin antibody (Sigma, Catalog No. M4276); 1:200 dilution of anti-Ki67 antibody (Abcam, Catalog No. ab16667). Detection was conducted using 1:1,000 dilution of anti-mouse IgG fragment, Alexa Fluor 555 conjugate (CST, Catalog No. #4409), and 1:1,000 dilution of anti-rabbit fragment, Alexa Fluor 555 conjugate (CST, Catalog No. #4413S). Nucleus was stained by DAPI in a final concentration of 0.1 μg/ml (CST, Catalog No. #4083). TUNEL assay was performed according to the manufacturer's instructions (MBL, Catalog No. 8445). Quantification of Ki67-positive rate and TUNEL-positive rate was conducted using ImageJ (version 1.51) software.

### Transmission electron microscopy

Transmission electron microscope analysis was performed on the gastrocnemius in standard fashion. Ultra-thin sections were stained with uranyl acetate and lead citrate, and then examined using a Tecnai transmission electron microscope (FEI) operated at 80 kV.

### Mouse embryos in situ hybridization

In situ hybridization using RNAscope probes was performed on E10.5 mouse embryos. Embryos were fixed with 4% paraformalde-hyde for 24 h at 4°C, dehydrated, and embedded in paraffin. Tissue sections were washed twice with PBS for 5 min, followed by incubation in hydrogen peroxide (ACD, Catalog No. 322335) for 10 min at room temperature, boiling in target retrieval (ACD, Catalog No. 322000) for 15 min. After target retrieval, slides were briefly washed with distilled water and incubated for 30 min at 40°C with Protease Plus (ACD, Catalog No. 322331). Following all pretreatments, the manufacturer's protocol for RNAscope 2.5 HD Detection Kit-Brown (ACD, Catalog No. 322310) was followed to hybridize probes and detect the signals. The following probes were used: RNAscope Probe-Mm-Pax3 (ACD, Catalog No. 455801); RNAscope Probe-Mm-Met (ACD, Catalog No. 405301); RNAscope Negative Control Probe-DapB (ACD, Catalog No. 310043); and

**The paper explained**

**Problem**

Arthrogryposis is a group of disorders characterized by congenital contractures of two or more joints, and it affects about 1/3,000 newborns. As a group of phenotypically and genetically heterogeneous disorders, the underlying molecular etiology and mechanism remain unknown in a large number of cases of arthrogryposis.

**Results**

In the present study, MET p.Y1234C mutation was found to be responsible for arthrogryposis in a four-generation pedigree. In vitro study showed that MET p.Y1234C mutation caused dysfunction of the phosphorylation and tyrosine kinase activity of MET protein. A Met p.Y1232C mutant (corresponding to MET p.Y1234C mutation in human beings) mouse model was constructed, and the phenotype of homozygotes was identical to Met null mutant mice, including embryonic lethality and complete loss of muscles that originate from migratory precursors. Heterozygous mice were born alive and showed reduction of myofibers in both appendicular and axial muscles. Further study demonstrated that the defective migration of muscle progenitor cells and impaired proliferation of secondary myoblasts were responsible for the muscular dysplasia in mutant mice.

**Impact**

This is the first study to report MET to be a disease-causing gene of arthrogryposis. We showed that arthrogryposis could be caused by the reduction of myofibers around the affected joints. Our study is also first to demonstrate a direct relationship between MET mutation and skeletal muscle dysplasia in human beings.

RNAscope Positive Control Probe-Mm-Ppib (ACD, Catalog No. 313911).

## Statistical analysis

All quantitative data are here presented as mean ± standard deviation (SD). Statistical analysis of body weight of transgenic mice was performed using one-way ANOVA followed by Dunnett's *post hoc* test for multiple comparisons. The positive rates of Ki67- and TUNEL-labeled cells were analyzed using the chi-square test ($\chi^2$ test). All statistical analyses were conducted with the SPSS (version 19.0) statistical software. The level of statistical significance was set at $P < 0.05$. See Appendix Statistical Analysis for the exact $P$-value in each experiment.

## Study approval

Written informed consent was obtained from all subjects or, in the case of children under 16 years of age, their parents. Collection and usage of patient samples for this study were approved by the Ethics Committee of the First Affiliated Hospital of Sun Yat-sen University. All procedures in studies involving human participants were performed in accordance with the principles set out in the WMA Declaration of Helsinki and the Department of Health and Human Services Belmont Report. Transgenic mice were raised in the laboratory animal center of Sun Yat-sen University, and all animal experiments, housing, and husbandry followed the operating procedures approved by the Institutional Animal Care and Use Committee of Sun Yat-sen University.

## Data availability

The whole-exome sequencing data of arthrogryposis patients are available in the Clinvar dataset (https://www.ncbi.nlm.nih.gov/clinvar/), and accession ID is SCV000606865.

**Expanded View** for this article is available online.

## Acknowledgements

We would like to express our gratitude to patients and families who participated in this study. This research was supported by the National Natural Science Foundation of China (No. 81772293, 81572091, 81772302, 81472039), General Financial Grand from the China Postdoctoral Science Foundation (No. 2017M622873), Guangzhou Science and Technology Project (No. 201704020120), and the First Affiliated Hospital of Sun Yat-sen University "Three in Three" Project.

## Author contributions

PS designed the experiments. HZ, CL, XY, SZ, ZL, TZ, CX, and XH conducted the experiments. TW, DS, XQ, XW, YL, BG, and YC raised transgenic mice. HZ and CL analyzed the data and wrote the manuscript. DH, GY, QF, and CG helped with the replication studies.

## Conflict of interest

The authors declare that they have no conflict of interest.

## For more information

The URLs for data presented in this article are as follows:

(i)     OMIM, http://www.omim.org/
(ii)    dbSNP, http://www.ncbi.nlm.nih.gov/projects/SNP/
(iii)   SeattleSeq Annotation 150, http://snp.gs.washington.edu/
(iv)    NCBI Build 37, hg19, http://genome.ucsc.edu/
(v)     1000 genomes, http://www.1000genomes.org/
(vi)    Mutation Taster, http://www.mutationtaster.org
(vii)   Polyphen-2, http://genetics.bwh.harvard.edu/pph2
(viii)  SIFT, http://sift.jcvi.org.
(ix)    Clinvar dataset, https://www.ncbi.nlm.nih.gov/clinvar/

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
