## [Review Process File · EMBO Molecular Medicine]

MET Mutation Causes Muscular Dysplasia and Arthrogryposis

Hang Zhou, Chengjie Lian, Tingting Wang, Xiaoming Yang, Caixia Xu, Deying Su, Shuhui Zheng, Xiangyu Huang, Zhiheng Liao, Taifeng Zhou, Xianjian Qiu, Yuyu Chen, Bo Gao, Yongyong Li, Xudong Wang, Guoling You, Qihua Fu, Christina Gurnett, Dongsheng Huang, Peiqiang Su

Review timeline:

Submission date:	21 August 2018
Editorial Decision:	22 September 2018
Revision received:	14 December 2018
Editorial Decision:	17 January 2019
Revision received:	28 January 2019
Accepted:	29 January 2019

Editor: Lise Roth

Transaction Report:

1st Editorial Decision

22 September 2018

Thank you for the submission of your manuscript to EMBO Molecular Medicine. We have now received feedback from two of the three reviewers who agreed to evaluate your manuscript. Given that referee 1 will unfortunately not be able to return his/her report in a timely manner, and that both referees 2 and 3 are overall positive, we prefer to make a decision now in order to avoid further delay in the process.

Addressing the referees' comments in full will be necessary for further considering the manuscript in our journal. Particular attention should be given to the discussion and to the quality of some of the pictures. You may know that EMBO Molecular Medicine encourages a single round of revision only and therefore, acceptance or rejection of the manuscript will depend on the completeness of your responses included in the next, final version of the manuscript.

EMBO Molecular Medicine has a "scooping protection" policy, whereby similar findings that are published by others during review or revision are not a criterion for rejection. Should you decide to submit a revised version, I do ask that you get in touch after three months if you have not completed it, to update us on the status. Please also contact us as soon as possible if similar work is published elsewhere. If other work is published, we may not be able to extend the revision period beyond three months.

I look forward to receiving your revised manuscript.

***** Reviewer's comments *****

Referee #2 (Remarks for Author):

This work identifies in a four generation Arthrogryposis pedigree with campodactyly a point

mutation in the Met receptor gene which appears to be causal in heterozygosity. The mutation converts into cysteine one of two tyrosine residues previously known to be essential for the kinase activity of the Met receptor. The authors corroborate their genetic data with biochemical work confirming in transfected cells that the mutation results in loss of the kinase activity of the receptor. They then describe a CRISP/Cas9 genetically modified mouse bearing the same loss of function mutation. As expected, in homozygosity the mutation was lethal and recapitulated the defects of a met null, including the complete lack of skeletal muscles of migratory origin, among which are those of the limbs. Interestingly, in the heterozygous state, the mutant mice showed an hypomorphic limb muscle phenotype, similar to the skeletal muscle dysplasia seen in the human patients. The authors conclude that in their pedigree arthrogryposis with campodactyly is likely to be due to reduction of the myofibers around the joints, reflecting the defective migration of myoblast precursors during embryogenesis. This in turn would be due to a partial loss of function of the myoblasts Met receptor, caused by expression of the inactive product of the mutated MET allele. The genetic and phenotypic analyses described in this ms are appropriate and the conclusion is convincing. As stated by the authors, this work is the first one to link a loss of function MET mutation to a skeletal muscle defect in humans. Thus, given the novelty of the observation and the evidence provided, the work merits publication. However I would require revising the ms according to the points outlined below.

1) In the Arthrogryposis pedigree and in the animal model the limb muscle phenotype is linked to a loss of function allele present in heterozygosity. Since previous work on met knock out mice or on mice mutated in the tyrosine residues involved in signal transduction did not show any phenotype in heterozygosity, in the present case (mutation inhibiting the kinase activity) the skeletal muscle defect is likely to be due to a dominant negative effect (ligand binding induces receptor dimerization) rather than to haploinsufficiency. A comment on this concept should be provided in the Discussion.

2) In the experiments aimed at showing the detrimental effect of the Y1234 mutation on ligand-induced receptor phosphorylation (Supplementary Fig.3) mouse HGF was used to stimulate human Met in transfected cells. It is known that mouse HGF is very inefficient in human Met stimulation (Rong et al., Mol Cell Biol 1992). The quality of these results could be improved by using commercially available human HGF.

3) The Results section on the effect of the Met mutation in secondary myogenesis (involving also axial muscles which do not seem to be affected in the patients) essentially recapitulates the original observations of Maina et al. (Cell, 1996) made in animal models of loss of Met transductional function. A reference to this work should be included at this point.

4) In the Discussion the authors make the point that a lot more is known in humans on the effect of gain of function MET mutations (in tumorigenesis) rather than on the consequences of MET loss of function genetic lesions. However there is a significant amount of work linking SNV causing lowered Met expression in neural tissue to Autistic Spectrum Disorders. Could the authors specify in the Discussion if there was any evidence of increased ASD susceptibility in this large pedigree?

Referee #3 (Remarks for Author):

This paper uses a 4 generation family with muscular dysplasia and arthrogryposis to identify the genetic etiology of these two conditions. Using whole-exome sequencing of members of this family they identify a MET c.A3701G mutation (that leads to pTyr1234Cys and loss of activation of MET tyrosine kinase activity) as the genetic cause of these phenotypes. They then generated a mouse with this same mutation (a mutation in this region had not previously been generated). Strikingly, the mice heterozygous for this mutation phenocopy many of the muscle phenotypes seen in the human carriers of this mutation - particularly the loss of palmar muscles, but normal development of foot muscles. However, unlike the human carriers, the mice did not exhibit arthrogryposis. The authors further examine the cause of the loss muscle in their mouse mutant. As expected from previous literature, they find migration of muscle progenitors into the limb and diaphragm is strongly affected by the MET mutations. In addition, confirming previous work in the mouse, in epaxial muscles (which do not require migration of muscle progenitors) they find that primary myogenesis is unaffected, but during secondary myogenesis proliferation of myoblasts is strongly affected by the

MET mutation.

Overall, this is a solid paper. I was impressed by the generation and analysis of the mice with the human mutation. Such experiments are not frequently done, but of course are the best evidence that a particular mutation underlies the etiology of human disease. Here they definitively show that the MET c.A3701G mutation (encoding a tyrosine known from previous biochemistry experiments to be critical for MET's function) underlies the muscle defects and confirm previously identified mechanisms by which MET regulates muscle development. Still unexplained is how mechanistically MET mutations give rise to arthrogryposis.

Below are some comments to improve the manuscript.

1. Supplemental Figure 1. The lower two panels showing longitudinal sections are unhelpful and should be removed.
2. Figure 1 D. It is difficult to compare the hindlimb muscles. Could the authors provide sections of WT and heterozygous that are better matched in terms of muscle anatomy?
3. The authors use the term "chondriosomes" on p. 13 line 14 and 16 and Figure legend for Fig. 2 i. I think they mean mitochondria.
4. Figure 3b and d. Please show the dermomyotome in the standard orientation where the neural tube is at the top of the figure and the dermomyotome will then be orientated approximately vertically.
5. Figures 3 c and d. It is nearly impossible to see the Met probe (in brown). Please fix this.
6. There are currently only 3 Figures and 6 Supplemental Figures. Some of the Supplemental Figures (particularly Supplemental Figure 3 and 6) could be moved to a Figure in the main part of the paper.
7. The authors state in the Discussion "among various Met loss-of-function mouse models, despite differences in the severity of the phenotypes, two common features were shared across all mouse models: the reduction of muscle fibers were more pronounced in forelimbs than in hindlimbs; the reduction of myofibers in the dorsal side of forelimbs was more severe than on the ventral side". I think the statement needs more specific support.
 - A. In the mouse mutant they created the only difference between fore and hindlimbs they report is in the complete absence of intrinsic muscles in the hand and the presence of intrinsic muscles in the foot. They also report no differences between effects of Met loss on dorsal versus ventral muscles. Did they examine this?
 - B. For mouse mutants that others have generated, they need to cite specific examples of differential effects of Met loss on fore versus hindlimb and dorsal versus ventral muscles.
8. The authors need to more explicitly acknowledge throughout the Results and Discussion that MET has previously been demonstrated to be required for migration of muscle progenitors into the limb, tongue and diaphragm and also is required for secondary myogenesis, and this is why they looked at these particular phenotypes in their MET mutant mice.
7. While the Met mutation in mouse phenocopies the muscle phenotypes in the human patients, the mouse mutants do not have the arthrogryposis characteristic of the patients. While the authors readily acknowledge this difference between mouse and humans, a bit more discussion of this would be helpful. For instance, are there any mouse mutants which have arthrogryposis? Is this a phenotype that just is not seen in mice? Other Met mutants have ventral flexion of the paws. Perhaps arthrogryposis does not happen in mice because the digits are relatively short so instead of particular digits flexing the whole paw flexes at the wrist.
8. Please correct the last sentence of the abstract "Overall, our study shows MET to be the causative

gene of arthrogryposis..." to "Overall, our study shows MET to be a causative gene of arthrogryposis..." As the authors readily acknowledge arthrogryposis is genetically heterogeneous.

1st Revision - authors' response

14 December 2018

Reviewer #2 (Remarks for Author)

Question 1: In the Arthrogryposis pedigree and in the animal model the limb muscle phenotype is linked to a loss of function allele present in heterozygosity. Since previous work on met knock out mice or on mice mutated in the tyrosine residues involved in signal transduction did not show any phenotype in heterozygosity, in the present case (mutation inhibiting the kinase activity) the skeletal muscle defect is likely to be due to a dominant negative effect (ligand binding induces receptor dimerization) rather than to haploinsufficiency. A comment on this concept should be provided in the Discussion.

Response: Thank you for your comments. In our study, heterozygous *Met* p.Y1232C mutant mice showed reduction in the number of myofibers in both appendicular and axial muscles. Given that various heterozygous *Met* loss-of-function mutant mice did not show any abnormal phenotypes (Maina et al, 1996; Maina et al, 2001; Sachs et al, 2000), the phenotype of *Met* p.Y1232C heterozygotes is likely to be caused by dominant negative effect instead of haploinsufficiency. Because Y-1232 is a crucial phosphorylation site in the mouse MET kinase domain, *Met* p.Y1232C mutation is likely to cause a dominant negative effect in a way similar to kinases-domain-truncated mutant MET described previously (Furge et al, 2001; Long et al, 2003). Because tyrosine kinase activation of MET depends on the dimerization of MET in response to ligand binding (Trusolino et al, 2010b), p.Y1234C mutant MET protein might form non-functional dimers with wild-type MET protein, resulting in impaired HGF-MET signal transduction. As shown in Fig. 1A for reviewers and editors only, co-expression of wild-type and mutant MET significantly down-regulated the phosphorylation of ERK1/2 and AKT, which are major downstream transducers of HGF-MET signal (Trusolino et al, 2010a). Moreover, co-expression of wild-type and mutant MET impaired the transcription of the target genes of HGF-MET signal (Fig. 1B for reviewers and editors only), indicating that *MET* p.Y1234C mutation may induce dominant negative effect *in vitro*.

These opinions about the dominant negative effect caused by *MET* p.Y1234C mutation have been added to the discussion section (Page 12, line 15-16 and Page 13, line 1-10) of the revised manuscript.

Reference:

- Furge KA, Kiewlich D, Le P, Vo MN, Faure M, Howlett AR, Lipson KE, Vande Woude GF, Webb CP (2001) Suppression of Ras-mediated tumorigenicity and metastasis through inhibition of the Met receptor tyrosine kinase. *Proc Natl Acad Sci USA* 98: 10722-7
- Long IS, Han K, Li M, Shirasawa S, Sasazuki T, Johnston M, Tsao MS (2003) Met receptor overexpression and oncogenic Ki-ras mutation cooperate to enhance tumorigenicity of colon cancer cells *in vivo*. *Mol Cancer Res* 1: 393-401
- Maina F, Casagrande F, Audero E, Simeone A, Comoglio PM, Klein R, Ponzetto C (1996) Uncoupling of Grb2 from the Met receptor *in vivo* reveals complex roles in muscle development. *Cell* 87: 531-42
- Maina F, Pante G, Helmbacher F, Andres R, Porthin A, Davies AM, Ponzetto C, Klein R (2001) Coupling Met to specific pathways results in distinct developmental outcomes. *Mol Cell* 7: 1293-306
- Sachs M, Brohmann H, Zechner D, Muller T, Hulsken J, Walther I, Schaeper U, Birchmeier C, Birchmeier W (2000) Essential role of Gab1 for signaling by the c-Met receptor *in vivo*. *J Cell Biol* 150: 1375-1384
- Trusolino L, Bertotti A, Comoglio PM (2010a) MET signalling: principles and functions in development, organ regeneration and cancer. *Nat Rev Mol Cell Biol* 11: 834-48
- Trusolino L, Bertotti A, Comoglio PM (2010b) MET signalling: principles and functions in development, organ regeneration and cancer. *Nat Rev Mol Cell Bio* 11: 834-848

Question 2: In the experiments aimed at showing the detrimental effect of the Y1234 mutation on ligand-induced receptor phosphorylation (Supplementary Fig.3) mouse HGF was used to stimulate human Met in transfected cells. It is known that mouse HGF is very inefficient in human Met

stimulation (Rong et al., Mol Cell Biol 1992). The quality of these results could be improved by using commercially available human HGF.

Response: Thank you for your comments. Per reviewer's advice, human HGF (PeproTch, catalog No. 100-39H-25) was used to stimulate human MET in 293T cells. As shown in Fig EV2 B-D, HGF treatment could not phosphorylate Y-1234/1235 (B), Y-1349 (C), and Y1356 (D) of mutant MET receptor, suggesting *MET* mutation impaired the activation of MET receptor. Moreover, the tyrosine kinase activity of mutant MET receptor was lost as well (Fig 1J).

Question 3: The Results section on the effect of the Met mutation in secondary myogenesis (involving also axial muscles which do not seem to be affected in the patients) essentially recapitulates the original observations of Maina et al. (Cell, 1996) made in animal models of loss of Met transductional function. A reference to this work should be included at this point.

Response: Thank you for your comments. The reference mentioned by the reviewer has been added in the introduction (Page 4, line 11) and discussion sections (Page 12, line 13) of the revised manuscript.

Question 4: In the Discussion the authors make the point that a lot more is known in humans on the effect of gain of function MET mutations (in tumorigenesis) rather than on the consequences of MET loss of function genetic lesions. However, there is a significant amount of work linking SNV causing lowered Met expression in neural tissue to Autistic Spectrum Disorders. Could the authors specify in the Discussion if there was any evidence of increased ASD susceptibility in this large pedigree?

Response: We would like to thank the reviewer for raising such an important issue. Amount of studies reported that SNVs on *MET* gene increased the susceptibility to Autistic Spectrum Disorders (ASD) (Peng et al, 2013; Plummer et al, 2013). A strong association between rs1858830 (located in the 5'-transcriptional regulatory region of the *MET* gene) and ASD has been reported. Functional assays revealed that rs1858830 can cause a decrease in *MET* promoter activity, and then down-regulate the mRNA and protein expression of MET (Campbell et al, 2006). To determine whether *MET* p.Y1234C mutation increases ASD susceptibility, patients in our pedigree were recruited and underwent the mental evaluation by a psychiatrist. They turned out to be mentally healthy, and no sign of ASD was observed.

Unlike Arthrogryposis, which is a monogenic disease and genetic disorder is the main cause of the disease, ASD is polygenic disease (Hall, 2014; Peng et al, 2013). In addition to genetic factors, epigenetic and environmental factors also play vital roles in its onset. Most of the studies that reported the association between SNVs on *MET* and ASD was based on the data from European or North American populations, while our arthrogryposis patients belong to the Chinese Hans population. The differences in genetic background may account for the inconsistency.

These opinions have been incorporated into the revised manuscript (Page 5, line 13-14 and Page 15, line 5-9).

Reference:

Peng Y, Huentelman M, Smith C, Qiu S (2013) MET receptor tyrosine kinase as an autism genetic risk factor. *Int Rev Neurobiol* 113: 135-65
 Plummer JT, Evgrafov OV, Bergman MY, Friez M, Haiman CA, Levitt P, Aldinger KA (2013) Transcriptional regulation of the MET receptor tyrosine kinase gene by MeCP2 and sex-specific expression in autism and Rett syndrome. *Transl Psychiatry* 3: e316
 Campbell DB, Sutcliffe JS, Ebert PJ, Militerni R, Bravaccio C, Trillo S, Elia M, Schneider C, Melmed R, Sacco R, Persico AM, Levitt P (2006) A genetic variant that disrupts MET transcription is associated with autism. *Proc Natl Acad Sci U S A* 103: 16834-9
 Hall JG (2014) Arthrogryposis (multiple congenital contractures): diagnostic approach to etiology, classification, genetics, and general principles. *Eur J Med Genet* 57: 464-72

Reviewer #3 (Remarks for Author):

Question 1. Supplemental Figure 1. The lower two panels showing longitudinal sections are unhelpful and should be removed.

Response: Thank you for your comments. We have deleted the lower two panels of previous Supplemental Figure 1a (Fig EV1A in the revised manuscript).

Question 2. Figure 1 D. It is difficult to compare the hindlimb muscles. Could the authors provide sections of WT and heterozygous that are better matched in terms of muscle anatomy?

Response: Thank you for your comments. Better matched images are now presented in Figure 2E of the revised manuscript.

Question 3. The authors use the term "chondriosomes" on p. 13 line 14 and 16 and Figure legend for Fig. 2 i. I think they mean mitochondria.

Response: Thank you for your comments, and the "chordrisomes" has been replaced by "mitochondria" in the revised manuscript (Page 34, line 7).

Question 4. Figure 3b and d. Please show the dermomyotome in the standard orientation where the neural tube is at the top of the figure and the dermomyotome will then be orientated approximately vertically.

Response: Thank you for your comments. We have revised these figures (Figure 3A and Fig EV3 in the revised manuscript) accordingly.

Question 5. Figures 3 c and d. It is nearly impossible to see the Met probe (in brown). Please fix this.

Response: Thank you for your comments, we have re-performed the *in situ* hybridization assay. In Fig EV3, the signal of Met probe is now clearly visible.

Question 6. There are currently only 3 Figures and 6 Supplemental Figures. Some of the Supplemental Figures (particularly Supplemental Figure 3 and 6) could be moved to a Figure in the main part of the paper.

Response: Thank you for your comments. Since we submitted this study as a Report, there is a limit to the total number of figures in the manuscript (3 at most). Per the reviewer's advice, we have re-organized all the figures, and we entered the major data into the main figures.

Question 7. The authors state in the Discussion "among various Met loss-of-function mouse models, despite differences in the severity of the phenotypes, two common features were shared across all mouse models: the reduction of muscle fibers were more pronounced in forelimbs than in hindlimbs; the reduction of myofibers in the dorsal side of forelimbs was more severe than on the ventral side". I think the statement needs more specific support.

A. In the mouse mutant they created the only difference between fore and hindlimbs they report is in the complete absence of intrinsic muscles in the hand and the presence of intrinsic muscles in the foot. They also report no differences between effects of Met loss on dorsal versus ventral muscles. Did they examine this?

B. For mouse mutants that others have generated, they need to cite specific examples of differential effects of Met loss on fore versus hindlimb and dorsal versus ventral muscles.

Response: Thanks so much for the valuable suggestions, and we have revised the discussion section per reviewer's advice (Page 13, line11-16; Page 14, line1-11).

In P0 heterozygous mice, there was a complete absence of the intrinsic muscles of the hand, while the intrinsic muscles in the foot remained normal. This inconsistency was similar to that observed in our arthrogryposis patients, who showed arthrogryposis and muscular dysplasia only on the upper limbs. One possible explanation for such inconsistency is that MET plays a predominant role in the early period of embryonic muscle development. As reported, MET was indispensable to the delamination of muscle progenitor cells from dermomyotome at the very beginning of migration. In *Met*^{-/-} mice, the muscle progenitor cells failed to delaminate from the dermomyotome, and all the muscles derived from migrational precursor cells failed to form as a result (Schmidt et al, 1995). Other genes, such as *LBX1* and *PAX3*, are more important in regulating muscle precursor's migration into the limbs during later period of migration (Masselink et al, 2017; Relaix et al, 2004). In *Lbx1*^{-/-} mice, limb muscle precursor cells could delaminate out of dermomyotome normally but failed to migrate into the limb, which led to the loss of appendicular muscles (a complete loss of muscles in hindlimbs and extensor muscles in forelimbs) (Gross et al, 2000). Forelimb buds (E9.0–E9.5) form earlier than hindlimb buds (E9.5–E10) during embryonic muscle development. It can be inferred that *Met* mutation causes more severe phenotypes in forelimbs than hindlimbs because it mainly affects the early stage of migration.

In *Met* loss-of-function mutant mice models constructed before, the reduction of myofibers in the extensor side of forelimbs was more severe than that in the flexor side. In mice with homozygous null mutation of *Gab1*, extensor muscles of the forelimbs were absent while the flexor muscles are normal (Sachs et al, 2000). In *Met* p.D1358H homozygous mice, the muscle reduction was more severe in the extensor side, which caused hyperflexed wrist at birth (Maina et al, 1996). A possible explanation for such inconsistency is that, different intensity of HGF-MET signal is required for the migration of muscle progenitor cells to dorsal or ventral forelimb. The dorsal forelimb may require stronger HGF-MET signal intensity, so the influence on migration is more severe on the dorsal forelimb in *Met* mutant mice.

Reference :

- Gross MK, Moran-Rivard L, Velasquez T, Nakatsu MN, Jagla K, Goulding M (2000) Lbx1 is required for muscle precursor migration along a lateral pathway into the limb. *Development* 127: 413-424
- Maina F, Casagrande F, Audero E, Simeone A, Comoglio PM, Klein R, Ponzetto C (1996) Uncoupling of Grb2 from the Met receptor in vivo reveals complex roles in muscle development. *Cell* 87: 531-42
- Masselink W, Masaki M, Sieiro D, Marcelle C, Currie PD (2017) Phosphorylation of Lbx1 controls lateral myoblast migration into the limb. *Dev Biol* 430: 302-309
- Relaix F, Rocancourt D, Mansouri A, Buckingham M (2004) Divergent functions of murine Pax3 and Pax7 in limb muscle development. *Gene Dev* 18: 1088-1105
- Sachs M, Brohmann H, Zechner D, Muller T, Hulsken J, Walther I, Schaeper U, Birchmeier C, Birchmeier W (2000) Essential role of Gab1 for signaling by the c-Met receptor in vivo. *J Cell Biol* 150: 1375-1384
- Schmidt C, Bladt F, Goedecke S, Brinkmann V, Zschiesche W, Sharpe M, Gherardi E, Birchmeier C (1995) Scatter factor/hepatocyte growth factor is essential for liver development. *Nature* 373: 699-702

Question 8. The authors need to more explicitly acknowledge throughout the Results and Discussion that MET has previously been demonstrated to be required for migration of muscle progenitors into the limb, tongue and diaphragm and also is required for secondary myogenesis, and this is why they looked at these particular phenotypes in their MET mutant mice.

Response: Thanks for your comments. We have revised the introduction and discussion sections of the manuscript accordingly (Page 4, line 8-11; Page 12, line 11-13).

Question 9. While the *Met* mutation in mouse phenocopies the muscle phenotypes in the human patients, the mouse mutants do not have the arthrogryposis characteristic of the patients. While the authors readily acknowledge this difference between mouse and humans, a bit more discussion of this would be helpful. For instance, are there any mouse mutants which have arthrogryposis? Is this a phenotype that just is not seen in mice? Other *Met* mutants have ventral flexion of the paws. Perhaps arthrogryposis does not happen in mice because the digits are relatively short so instead of particular digits flexing the whole paw flexes at the wrist.

Response: Thanks for the valuable suggestions. The reason for the absence of arthrogryposis characteristic in the *Met* p.Y1232C mutant mice is now discussed in the revised manuscript (Page 14, line 11-16; Page 15, line 1-4).

It is also noteworthy that although *Met* p.Y1232C mutant mice recapitulated the phenotypes of muscular dysplasia in arthrogryposis patients, neither heterozygous nor homozygous mice showed contracture of distal joint. In arthrogryposis family we collected, all patients with *MET* p.Y1234C mutation had camptodactyly, which was caused by the loss of intrinsic muscles of the hands. Similar to these patients, *Met* c.A3695G mutant mice also showed loss of intrinsic muscles in the hands. The reason why mutant mice had no camptodactyly may be that the digits of mice are relatively short, so contracture is not readily visible. However, instead of contracture of digits, the whole paw flexion at the wrist could be observed in *Met* p.Y1356F mutant homozygotes (Maina et al, 1996). In *Met* p.Y1356F mice, the reduction of myofibers in the extensor side of forelimbs was significantly severer than that in the flexor side, thereby causing the hyperflexed forelimbs. In *Met* p.Y1232C mice, there were no obvious differences in reduction of myofibers in extensor and flexor sides of forelimbs, so no flexion of paws was observed.

Reference :

- Maina F, Casagrande F, Audero E, Simeone A, Comoglio PM, Klein R, Ponzetto C (1996)

Uncoupling of Grb2 from the Met receptor in vivo reveals complex roles in muscle development.
Cell 87: 531-42

Question 10. Please correct the last sentence of the abstract "Overall, our study shows MET to be the causative gene of arthrogryposis..." to "Overall, our study shows MET to be a causative gene of arthrogryposis..." As the authors readily acknowledge arthrogryposis is genetically heterogeneous.
Response: Thanks for your comment. The corresponding sentence in abstract has been corrected (Page 2, line 16; Page15, line 10).

2nd Editorial Decision

17 January 2019

Thank you for the submission of your revised manuscript to EMBO Molecular Medicine, and please accept my apologies for the delay in getting back to you, which is due the fact that the manuscript was sent back to the reviewers just before the holidays break. We have now received the enclosed reports from the two referees who were asked to re-assess it. As you will see, the reviewers are now supportive, and I am pleased to inform you that we will be able to accept your manuscript pending the following final editorial amendments:

1) Referees' comments:

Please address the minor text changes and discussion points raised by the referees.

***** Reviewer's comments *****

Referee #2 (Remarks for Author):

In the new version of the manuscript the authors satisfied all my requests. However in revising the manuscript the Authors had some problems of clarity. My suggestions to modify the text are attached below.

Page 4 - Lines 8-11

In skeletal muscle development, MET has been demonstrated not only to be crucial for the migration of muscle progenitor cells into the limbs, tongue, and diaphragm, but also to be necessary for the proliferation of secondary myoblasts in the trunk.

Page 5 - Lines 13-14 Move to the Discussion section, see below Page 16 Lines 5-9

Page 13 - Lines 1-10

...the phenotype of Met p.Y1232C heterozygotes is likely to be caused by a dominant negative effect rather than by haploinsufficiency. Since Y-1232 is a crucial phosphorylation site in the MET kinase domain and MET activation depends on receptor dimerization in response to ligand binding (Trusolino et al, 9 2010), the p.Y1232C mutant might form non-functional dimers with the wild-type MET protein, resulting in impaired HGF-MET signaling. A similar effect has been previously described with a kinase-domain-truncated mutant MET (Furge et al, 2001; Long et al, 2003).

Pag 16 - Lines 5-9

It has been previously reported that SNVs in the MET gene, causing lowered MET expression, increase susceptibility to Autistic Spectrum Disorders (ASD) in European and North American populations (Campbell et al, 2006). Mental evaluation by a psychiatrist of our arthrogryposis patients revealed no sign of ASD, possibly owing to the difference in genetic background among populations.

Referee #3 (Remarks for Author):

The authors have answered almost all my previous concerns. However, I have two small outstanding issues.

1. I previously asked the authors to explicitly discuss whether previous Met mutants showed a difference between fore and hind limb muscles. They answered this in the point-by-point discussion

to the reviewer, but I do not see this in the actual manuscript. This should be included in the manuscript.

2. The authors previously mentioned that previous analyses of Met mutants found differences between development of dorsal and ventral limb muscles. I asked the authors whether they had looked at this and to report whether or not they found differences. There is no mention in the revised manuscript of dorsal/ventral differences or whether they even looked. Did they not check this? If so, what was the result? This is not necessary for publication, but I was curious since they had mentioned this in the previous version of the manuscript.

Corresponding Author Name: Peiqiang Su

Manuscript Number: EMM-2018-09709-V2